# Second-Order Convergence in Private Stochastic Non-Convex Optimization

**Youming Tao**
TU Berlin & Shandong University
tao@ccs-labs.org

**Zuyuan Zhang**
The George Washington University
zuyuan.zhang@gwu.edu

**Dongxiao Yu**
Shandong University
dxyu@sdu.edu.cn

**Xiuzhen Cheng**
Shandong University
xzcheng@sdu.edu.cn

**Falko Dressler**
TU Berlin
dressler@ccs-labs.org

**Di Wang**
KAUST
di.wang@kaust.edu.sa

## Abstract

We investigate the problem of finding second-order stationary points (SOSP) in differentially private (DP) stochastic non-convex optimization. Existing methods suffer from two key limitations: **(i)** inaccurate convergence error rate due to overlooking gradient variance in the saddle point escape analysis, and **(ii)** dependence on auxiliary private model selection procedures for identifying DP-SOSP, which can significantly impair utility, particularly in distributed settings. To address these issues, we propose a generic perturbed stochastic gradient descent (PSGD) framework built upon Gaussian noise injection and general gradient oracles. A core innovation of our framework is using model drift distance to determine whether PSGD escapes saddle points, ensuring convergence to approximate local minima without relying on second-order information or additional DP-SOSP identification. By leveraging the adaptive DP-SPIDER estimator as a specific gradient oracle, we develop a new DP algorithm that rectifies the convergence error rates reported in prior work. We further extend this algorithm to distributed learning with heterogeneous data, providing the first formal guarantees for finding DP-SOSP in such settings. Our analysis also highlights the detrimental impacts of private selection procedures in distributed learning under high-dimensional models, underscoring the practical benefits of our design. Numerical experiments on real-world datasets validate the efficacy of our approach.

## 1 Introduction

Stochastic optimization is a fundamental problem in machine learning and statistics, aimed at training models that generalize well to unseen data using a finite sample drawn from an unknown distribution. As the volume of sensitive data continues to grow, privacy has become a pressing concern. This has led to the widespread adoption of differential privacy (DP) [11], which provides rigorous privacy guarantees while preserving model utility in learning tasks.

In the past decade, significant progress has been made in DP stochastic optimization, particularly for convex objectives [8, 29, 41, 39, 43]. While convex problems are relatively well understood, non-convex optimization introduces unique challenges, primarily due to the presence of saddle points.

39th Conference on Neural Information Processing Systems (NeurIPS 2025).

Most existing DP algorithms for non-convex problems focus on finding first-order stationary points (FOSP), characterized by small gradient norms [2, 5, 54]. However, FOSP include not only local minima but also saddle points and local maxima, often leading to suboptimal solutions [21, 42]. Consequently, second-order stationary points (SOSP), where the gradient is small and the Hessian is positive semi-definite, are more desirable as they guarantee convergence to local minima.

Motivated by this, substantial research has been devoted to finding SOSP in non-convex optimization [14, 24, 10, 22, 17]. However, the study of SOSP under differential privacy constraints (DP-SOSP) remains limited. At the same time, distributed learning has become increasingly important for training large-scale models across decentralized edge devices. Yet, no existing work has addressed DP-SOSP in non-convex stochastic optimization under distributed settings. Compared to single-machine setups, distributed learning introduces additional challenges, including data heterogeneity, cross-participant privacy, and communication efficiency.

**Limitations in the State-of-the-Art.** A notable exception in the study of DP-SOSP for stochastic optimization is the recent work by [30], which injects additional Gaussian noise into the DP gradient estimator near saddle points to facilitate escape. Despite its contributions, this method suffers from two key limitations. **(i)** Its saddle point escape analysis overlooks the variance of gradients, leading to incorrect error bounds. A direct correction of the analysis would unfortunately yield a weaker type of SOSP guarantee than originally targeted. This is because their design relies on additional injected noise beyond the inherent DP noise for escape, highlighting the need for an effective way of exploiting the DP noise already present. **(ii)** Their learning algorithm outputs all model iterates and guarantees only the *existence* of a DP-SOSP, requiring an auxiliary private model selection procedure to identify one. While effective in single-machine settings, it faces critical issues in distributed environments due to decentralized data access. In particular, auxiliary private selection introduces non-negligible error and communication overhead, especially when sharing high-dimensional second-order information. These drawbacks also underscore the necessity of a new learning algorithm that inherently outputs a DP-SOSP without dependence on any additional private selection procedure.

**Our Contributions.** We refer to Appendix A for more detailed discussions of the limitations outlined above. To address the challenges identified above, we propose a generic algorithmic and analytical framework for finding DP-SOSP in stochastic non-convex optimization. Our approach not only corrects existing error rates but also extends naturally to distributed learning. The main contributions are summarized as follows:

**1. A generic non-convex stochastic optimization framework:** We introduce a perturbed stochastic gradient descent (PSGD) framework that employs Gaussian noise and general stochastic gradient oracles. This framework serves as a versatile optimization tool for non-convex stochastic problems beyond the DP setting. A key innovation is a novel criterion based on model drift distance, which enables provable saddle point escape and guarantees convergence to approximate local minima with low iteration complexity and high probability.

**2. Corrected error rates for DP non-convex optimization:** By incorporating the adaptive DP-SPIDER estimator as the gradient oracle, we develop a differentially private algorithm that achieves a corrected error rate bound of $\tilde{O}\left(\frac{1}{n^{1/3}} + \left(\frac{\sqrt{d}}{\epsilon n}\right)^{2/5}\right)$, where $n$ is the number of samples. This corrects the suboptimal bound of $\tilde{O}\left(\frac{1}{n^{1/3}} + \left(\frac{\sqrt{d}}{\epsilon n}\right)^{3/7}\right)$ reported in [30].

**3. Application to distributed learning:** We extend the adaptive DP-SPIDER estimator to distributed learning. Via adaptivity, our learning algorithm improves upon the DIFF2 [37], which only guarantees convergence to DP-FOSP under *homogeneous* data. In contrast, our method provides the first error bound for converging to DP-SOSP under *heterogeneous* data: $\tilde{O}\left(\frac{1}{(mn)^{1/3}} + \left(\frac{\sqrt{d}}{\epsilon mn}\right)^{2/5}\right)$, where $m$ is the number of participants and $n$ is the number of samples per participant. Furthermore, we analyze the adverse effects of private model selection, showing that it deteriorates utility guarantees in high-dimensional regimes, thereby highlighting the necessity of our framework.

Due to the space limit, **technical lemmata**, **omitted proofs**, **experimental results** and **broader impacts**, **conclusions** are all included in the Appendix.

## 2   Related Work

**Private Stochastic Optimization.** Differential privacy (DP) has become a crucial consideration in stochastic optimization due to increasing concerns about data privacy. The pioneering work by [11] established the foundational principles of DP, and its application in stochastic optimization has since seen significant progress. Early efforts primarily focused on convex optimization, achieving strong privacy guarantees while ensuring efficient learning, with a long list of representative works e.g., [6, 51, 48, 4, 47, 49, 15, 5, 20, 43, 41, 8, 40]. Recent advances have extended DP to non-convex settings, mainly focusing on first-order stationary points (FOSP). Notable works in this area include [46, 54, 5, 52, 2], which improved error rates in non-convex optimization with balanced privacy and utility in stochastic gradient methods. However, these works generally fail to address the more stringent criterion of second-order stationary points (SOSP). The very recent work [30] tired to narrow this gap, but unfortunately has some issues in their results as we discussed before. Our work builds on this foundation by correcting error rates and proposing a framework that ensures convergence to SOSP while maintaining DP.

**Finding SOSP.** In non-convex optimization, convergence to FOSP is often insufficient, as saddle points can lead to sub-optimal solutions [21, 42]. Achieving SOSP, where the gradient is small and the Hessian is positive semi-definite, ensures that the optimization converges to a local minimum rather than a saddle point. Techniques for escaping saddle points, such as perturbed SGD with Gaussian noise, have been explored in works like [17] and [24]. [17] first showed that SGD with a simple parameter perturbation can escape saddle points efficiently. Later, the analysis was refined by [22, 24]. Recently, variance reduction techniques have been applied to second-order guaranteed methods [18, 28]. These methods ensure escape from saddle points by introducing noise to the gradient descent process. In contrast, the studies of SOSP under DP are quite limited, and most of them only consider the empirical risk minimization objective, such as [46, 50, 3]. Very recently, [30] addressed the population risk minimization objective, but with notable gaps in their error analysis, particularly in the treatment of gradient variance. Moreover, all of these works are limited to the single-machine setting and cannot be directly extended to the more general distributed learning setting.

**Distributed Learning.** With the rise of large-scale models and decentralized data, distributed learning has gained significant attention. Methods like federated learning [34] have enabled multiple clients to collaboratively train models without sharing their local data. Recent studies, such as [16, 32, 33] have investigated DP learning in distributed settings, but these works are limited to first-order optimality. While some studies have investigated SOSP in distributed learning, their focus was primarily on Byzantine-fault tolerance [53], and communication efficiency [36, 7]. No effort, to our knowledge, has been made to to ensure DP-SOSP in distributed learning scenarios with heterogeneous data. Our proposed framework fills this gap by introducing the first distributed learning algorithm with DP-SOSP guarantees while effectively handling data heterogeneity across clients.

## 3   Preliminaries

**Notations.** We denote by $\|\cdot\|$ the $\ell_2$ norm and by $\lambda_{\min}(\cdot)$ the smallest eigenvalue of a matrix. The symbol $\mathbf{I}_d$ represents the $d$-dimensional identity matrix. We use $O(\cdot)$ and $\Omega(\cdot)$ to hide constants independent of problem parameters, while $\tilde{O}(\cdot)$ and $\tilde{\Omega}(\cdot)$ additionally hide polylogarithmic factors.

**Stochastic Optimization.** Let $f : \mathbb{R}^d \times \mathcal{Z} \to \mathbb{R}$ be a (potentially non-convex) loss function, where $x \in \mathbb{R}^d$ denotes the $d$-dimensional model parameter and $z \in \mathcal{Z}$ is a data point.

**Assumption 1.** The loss function $f(\cdot; z)$ is $G$-Lipschitz, $M$-smooth, and $\rho$-Hessian Lipschitz. Specifically, for any $z \in \mathcal{Z}$ and any $x_1, x_2 \in \mathbb{R}^d$, we have: (i) $|f(x_1; z) - f(x_2; z)| \leq G\|x_1 - x_2\|$; (ii) $\|\nabla f(x_1; z) - \nabla f(x_2; z)\| \leq M\|x_1 - x_2\|$; (iii) $\|\nabla^2 f(x_1; z) - \nabla^2 f(x_2; z)\| \leq \rho\|x_1 - x_2\|$.

Let $\mathcal{D}$ denote the unknown data distribution. The population risk is defined as the *expected* loss: $F_{\mathcal{D}}(x) := \mathbb{E}_{z \sim \mathcal{D}}[f(x; z)]$ for $\forall x \in \mathbb{R}^d$. When clear from context, we omit $\mathcal{D}$ and simply write $F(x)$.

**Assumption 2.** Let $x^*$ denote a minimizer of the population risk and $F^* = F(x^*)$ its minimum value. There exists $U \in \mathbb{R}$ such that $\max_x F(x) - F^* \leq U$.

Let $D$ denote a dataset of $n$ i.i.d. samples from $\mathcal{D}$. The empirical risk is defined as $\hat{f}_D(x) := \frac{1}{|D|} \sum_{z \in D} f(x; z)$. Given access to $D$, the goal is to find an approximate second-order stationary point (SOSP) of the unknown population risk $F(\cdot)$. In general, we have the notion of $(\alpha_g, \alpha_H)$-SOSP:

**Definition 1** (($\alpha_g, \alpha_H$)-SOSP). A point $x$ is an $(\alpha_g, \alpha_H)$-SOSP of a twice differentiable function $F(\cdot)$ if $x$ satisfies $\|\nabla F(x)\| \leq \alpha_g$ and $\nabla^2 F(x) \succeq -\alpha_H \cdot \mathbf{I}_d$.

As shown in [53, Proposition 1], there exists a lower bound of $\tilde{O}(\alpha_g^{1/2})$ for $\alpha_H$ given $\alpha_g$, implying that an $(\alpha, \tilde{O}(\sqrt{\alpha}))$-SOSP is the best second-order guarantee achievable. Accordingly, we target the notion of $\alpha$-SOSP in this work, following [30].

**Definition 2** ($\alpha$-SOSP). A point $x$ is an $\alpha$-SOSP of a twice differentiable function $F(\cdot)$ if $x$ satisfies $\|\nabla F(x)\| \leq \alpha$ and $\nabla^2 F(x) \succeq -\sqrt{\rho\alpha} \cdot \mathbf{I}_d$.

An $\alpha$-SOSP excludes $\alpha$-strict saddle points where $\nabla^2 F(x) \preceq -\sqrt{\rho\alpha}\mathbf{I}_d$, thereby ensuring convergence to an approximate local minimum. Following prior work [30, 24], we assume $M \geq \sqrt{\rho\alpha}$ so that finding an SOSP is strictly more challenging than finding an FOSP.

**Distributed Learning.** In the distributed (federated) learning setting, $m$ clients collaboratively learn under the coordination of a central server. Each client $j \in [m]$ has a local dataset $D_j$ of size $n$, sampled from an unknown local distribution $\mathcal{D}_j$. The population risk for client $j$ is defined as $F_{\mathcal{D}_j}(x) := \mathbb{E}_{z \sim \mathcal{D}_j}[f(x; z)]$ or simply $F_j(x)$. The global population risk is defined as the average of the local population risks: $F_{\mathcal{D}}(x) := \frac{1}{m} \sum_{j \in [m]} F_j(x)$, or simply $F(x)$. We allow for heterogeneous local datasets, meaning that the local distributions $\{\mathcal{D}_j\}_{j \in [m]}$ may differ.

**Differential Privacy.** We aim to find an $\alpha$-SOSP under the requirment of Differential Privacy (DP), which is referred to as an $\alpha$-DP-SOSP. We say two datasets $D$ and $D'$ are *adjacent* if they differ by at most one record. DP ensures that the output of the stochastic optimization algorithm on any pair of adjacent datasets is statistically indistinguishable.

**Definition 3** (Differential Privacy (DP) [11]). Given $\epsilon, \delta > 0$, a randomized algorithm $\mathcal{A} : \mathcal{Z} \to \mathcal{X}$ is $(\epsilon, \delta)$-DP if for any pair of adjacent datasets $D, D' \subseteq \mathcal{Z}$, and any measurable subset $S \subseteq \mathcal{X}$,

$$\mathbb{P}[\mathcal{A}(D) \in S] \leq \exp(\epsilon) \cdot \mathbb{P}[\mathcal{A}(D') \in S] + \delta.$$

In distributed learning, we focus on *inter-client record-level DP (ICRL-DP)*, which assumes that clients do not trust the server or other clients with their sensitive local data. This notion has been widely adopted in state-of-the-art distributed learning works, such as [16, 32, 33].

**Definition 4** (Inter-Client Record-Level DP (ICRL-DP)). Given $\epsilon, \delta > 0$, a randomized algorithm $\mathcal{A} : \mathcal{Z}^m \to \mathcal{X}$ satisfies $(\epsilon, \delta)$-ICRL-DP if, for any client $j \in [m]$ and any pair of local datasets $D_j$ and $D_j'$, the full transcript of client $j$'s sent messages during the learning process satisfies (3), assuming fixed local datasets for other clients.

**Variance Reduction via SPIDER.** Since the population risk $F(\cdot)$ is unknown, standard SGD approximates the true gradient $\nabla F(x_{t-1})$ at iteration $t$ using a stochastic estimate $g_t$. However, such estimates often exhibit high variance, degrading convergence. The Stochastic Path Integrated Differential Estimator (SPIDER) [13] mitigates this variance using two gradient oracles $\mathcal{O}_1$ and $\mathcal{O}_2$. For a mini-batch $\mathcal{B}_t$ at iteration $t$, we define

$$\mathcal{O}_1(x_{t-1}, \mathcal{B}_t) := \nabla \hat{f}_{\mathcal{B}_t}(x_{t-1}), \quad \mathcal{O}_2(x_{t-1}, x_{t-2}, \mathcal{B}_t) := \nabla \hat{f}_{\mathcal{B}_t}(x_{t-1}) - \nabla \hat{f}_{\mathcal{B}_t}(x_{t-2}).$$

SPIDER queries $\mathcal{O}_1$ every $p$ iterations to refresh the gradient estimate. Between these updates, it uses $\mathcal{O}_2$ to incrementally refine the estimate:

$$g_t = \begin{cases} \mathcal{O}_1(x_{t-1}, \mathcal{B}_t), & \text{if } (t-1) \bmod p = 0, \\ g_{t-1} + \mathcal{O}_2(x_{t-1}, x_{t-2}, \mathcal{B}_t), & \text{otherwise.} \end{cases}$$

For smooth functions, the variance of $\mathcal{O}_2(x_{t-1}, x_{t-2}, \mathcal{B}_t)$ scales with $\|x_{t-1} - x_{t-2}\|$, which is typically small when updates are minimal. This allows SPIDER to achieve low-variance gradient estimates while maintaining accuracy.

We choose SPIDER because it achieves state-of-the-art error rates for privately finding first-order stationary points (DP-FOSP) [2]. Our goal is to investigate whether its variance reduction can extend to DP-SOSP. Importantly, the insights in this paper are not specific to SPIDER; they also apply to other variance-reduced methods such as STORM [9] or SARAH [38]. However, since these algorithms are conceptually similar, no significant improvement is expected from substituting them.

# 4 Our Generic Perturbed SGD Framework

In this section, we introduce a generic framework for finding an $\alpha$-SOSP of the population risk $F_{\mathcal{D}}(\cdot)$ by escaping saddle points. Our framework is a Gaussian perturbed stochastic gradient descent method, denoted as `Gauss-PSGD`.

## 4.1 Gradient Oracle Setup

Since $\nabla F_{\mathcal{D}}(\cdot)$ is unknown, direct gradient descent is infeasible. As in standard stochastic optimization, we assume access to a stochastic gradient oracle $g_t$ that approximates $\nabla F_{\mathcal{D}}(x_{t-1})$ at iteration $t$. For example, $g_t$ can be computed as an empirical gradient over a mini-batch $\mathcal{B}_t$ sampled from $\mathcal{D}$. We model the oracle as

$$g_t = \nabla F(x_{t-1}) + \zeta_t, \tag{1}$$

where $\zeta_t$ represents inherent gradient noise. Following [24, 30], we assume $\zeta_t \sim \mathrm{nSG}(\sigma)$, where nSG denotes a norm-sub-Gaussian distribution (Definition 7 in Appendix B).

To enable saddle point escape, we introduce an additional Gaussian perturbation to form a perturbed gradient oracle $\hat{g}_t$:

$$\hat{g}_t = g_t + \xi_t = \nabla F(x_{t-1}) + \zeta_t + \xi_t, \tag{2}$$

where $\xi_t \sim \mathcal{N}(0, r^2 \mathbf{I}_d)$. We define the effective noise magnitude in $\hat{g}_t$ as

$$\psi := \sqrt{\sigma^2 + r^2 d}. \tag{3}$$

The model update is then performed by

$$x_t \leftarrow x_{t-1} - \eta \hat{g}_t. \tag{4}$$

---

**Algorithm 1:** `Gauss-PSGD`: Gaussian Perturbed Stochastic Gradient Descent

---

**Input:** Failure probability $\omega$, initial model $x_0$, learning rate $\eta$, # of escape repeats $Q$, model deviation threshold $\mathcal{R}$, # of escape steps $\Gamma$

1   $t \leftarrow 0$;
2   **while** true **do**
3     $t \leftarrow t + 1$;
4     $\hat{g}_t \leftarrow$ `P_Grad_Oracle(*)`;
5     **if** $\|\hat{g}_t\| \leq 3\chi$ **then**
       /* Saddle point escape   */
6       $\tilde{t} \leftarrow t, \tilde{x} \leftarrow x_{t-1}, \mathsf{esc} \leftarrow \text{false}$;
7       **for** $q \leftarrow 1, \cdots, Q$ **do**
8         $t \leftarrow \tilde{t}, x_t \leftarrow \tilde{x}$;
9         **for** $\tau \leftarrow 1, \cdots, \Gamma$ **do**
10          $\hat{g}_t \leftarrow$ `P_Grad_Oracle(*)`;
11          $x_t \leftarrow x_{t-1} - \eta \cdot \hat{g}_t$;
12          **if** $\|x_t - \tilde{x}\| \geq \mathcal{R}$ **then**
13           $\mathsf{esc} \leftarrow \text{true}$;
14           **break**;
15          **else**
16           $t \leftarrow t + 1$;
17       **if** $\mathsf{esc} = \text{true}$ **then**
18        **break**;
19       **if** $\mathsf{esc} = \text{false}$ **then**
20        **return** $x_{t-1}$
21     **else**
       /* Normal descent step   */
22       $x_t \leftarrow x_{t-1} - \eta \cdot \hat{g}_t$;

---

Our problem setting fundamentally differs from that in [24]. In their setting, the target error $\alpha$ is given, and the perturbation magnitude $r$ is determined accordingly. In contrast, in our privacy-constrained setting, $r$ is dictated by the privacy parameters $(\epsilon, \delta)$, and the goal is to achieve the smallest possible $\alpha$ under this constraint. Crucially, their parameterization $r = O(\sqrt{(\sigma^2 + \alpha^{3/2})/d})$ implies that $r$ depends on both $\sigma$ and $\alpha$, determined by $\max\{\sigma/\sqrt{d}, \alpha^{3/4}/\sqrt{d}\}$. This non-invertible relationship between $r$ and $\alpha$ makes their setting incompatible with ours. First, under DP constraints, $r$ is determined by $(\epsilon, \delta)$ and may be smaller than $\sigma/\sqrt{d}$ in weak privacy regimes, violating the required lower bound. Second, because $r$ and $\alpha$ are not uniquely determined by each other, it is not meaningful to directly translate their error bounds into our setting. Thus, their analysis and results cannot be directly applied to our problem.

## 4.2 Our Approach: A General Gaussian-Perturbed SGD Framework

We present our `Gauss-PSGD` framework in Algorithm 1, which finds an $\alpha$-SOSP with high probability at least $1 - \omega$. As specified in (2), we employ a general Gaussian-perturbed stochastic gradient oracle, denoted as `P_Grad_Oracle(*)` in steps 4 and 10, where $*$ abstracts the specific arguments required by the oracle implementation. This abstraction allows `Gauss-PSGD` to serve as a flexible optimization framework for non-convex stochastic problems, applicable beyond the differential privacy (DP) setting.

At each iteration, the gradient estimate $\hat{g}_t$ is computed by `P_Grad_Oracle(*)`, and the model parameter is updated via the gradient descent step in (4). The algorithm proceeds until it encounters a

point $\tilde{x}$ satisfying $\|\hat{g}_t\| \leq 3\chi$, where $\chi$ is specified in (5). This point $\tilde{x}$ may lie near a saddle point with a large negative eigenvalue of the Hessian. To escape such a saddle point, the framework enters an escape procedure (steps 6–20), which performs $Q$ rounds of $\Gamma$-`descent` (steps 9–16).

In each round, the algorithm executes at most $\Gamma$ perturbed SGD iterations starting from $\tilde{x}$. If at any iteration we observe $\|x_t - \tilde{x}\| \geq \mathcal{R}$ for a threshold $\mathcal{R}$ (specified in (5)), indicating that the iterate has moved sufficiently far from $\tilde{x}$, we declare that the algorithm has successfully escaped the saddle point and resume normal PSGD from $x_t$. If no such movement is observed after $Q$ rounds, we declare $\tilde{x}$ an $\alpha$-SOSP of the population risk $F_{\mathcal{D}}(\cdot)$ and output $\tilde{x}$. The repetition over $Q$ rounds ensures a high probability of escape: as we will prove later, each $\Gamma$-`descent` succeeds in escaping a saddle point with constant probability, and multiple repetitions reduce the failure probability to any desired level.

A central innovation of our framework is using model drift distance as the escape criterion (step 12), replacing the function value decrease criterion used in [22, 24]. This design enables the algorithm to identify an SOSP with high probability during the optimization process itself, eliminating the need for an auxiliary private model selection step. Our key insight is as follows: escaping a saddle point not only causes a decrease in the objective function [22, 24] but also induces a substantial displacement of the model parameter beyond a threshold $\mathcal{R}$. Shifting from monitoring function values to tracking parameter movement is critical in population risk settings, where the objective function is unknown and function evaluations are unavailable, unlike in empirical risk minimization [22]. However, the model iterates and their deviations are observable. By leveraging this property, our framework can directly output an SOSP, rather than merely guaranteeing its existence among the iterates.

### 4.3 Main Results for `Gauss-PSGD` Framework

We begin by introducing the parameter setup and notations used throughout the analysis:

$$\iota := s\mu, \quad \chi := 4\sqrt{C}s\mu^2\psi, \quad \alpha := 4\chi,$$

$$\Gamma := \frac{\iota}{s\eta\sqrt{\rho\alpha}}, \quad \mathcal{R} := \frac{1}{\iota^{1.5}}\sqrt{\frac{\alpha}{\rho}}, \quad \Phi := \frac{s}{8\iota^3}\sqrt{\frac{\alpha^3}{\rho}}, \quad \eta := \frac{\sqrt{\rho\alpha}}{M^2\iota^2}. \tag{5}$$

where $s$ is a sufficiently large absolute constant to be chosen later, and $\mu$ is a logarithmic factor:

$$\mu := \max\left\{\frac{1}{s}\log\left(\frac{9d\log\left(\frac{4C^{1/4}}{s\eta r}\sqrt{\frac{\psi}{\rho}}\right)}{C^{1/4}\eta\sqrt{s\rho\psi}}\right), \log\left(\frac{160\sqrt{2}C^{1/4}}{s\sqrt{\eta r}}\sqrt{\frac{\psi}{\rho}}\right), \frac{\left(C\log\frac{4T}{\omega}\right)^{1/4}}{2^{\frac{3}{4}}\sqrt{s}}, 1\right\}. \tag{6}$$

Here $C$ is an absolute constant that may change across expressions. The rationale behind these parameter choices is further discussed in Remark 2 following Theorem 1. Let $\tilde{x}$ denote a saddle point of the population risk $F(\cdot)$, and $\mathcal{H} := \nabla^2 F(\tilde{x})$. Let $v_{\min}$ be the eigenvector corresponding to $\lambda_{\min}(\mathcal{H})$, and $\mathcal{P}_{-v_{\min}}$ be the projection onto the orthogonal complement of $v_{\min}$. Set $\gamma := -\lambda_{\min}(\mathcal{H})$.

**Definition 5** (Coupling Sequence). Let $\{x_i\}$ and $\{x_i'\}$ be two PSGD sequences initialized at $\tilde{x}$. We say they are *coupled* if they share the same randomness for $\mathcal{P}_{-v_{\min}}\xi_t$ and $\zeta_t$ at each iteration $t$, but use opposite perturbations in the $v_{\min}$ direction: $v_{\min}^\top\xi_t = -v_{\min}^\top\xi_t'$.

The following lemma ensures that under $\Gamma$-`descent`, at least one of the coupled sequences escapes the saddle point with constant probability (proof in Appendix C.1).

**Lemma 1** (Escaping Saddle Points). Let $\{x_i\}$ and $\{x_i'\}$ be coupled PSGD sequences initialized at $\tilde{x}$ such that $\|\nabla F(\tilde{x})\| \leq \alpha$ and $\lambda_{\min}(\nabla^2 F(\tilde{x})) \leq -\sqrt{\rho\alpha}$. Then, with probability at least $1/4$, there exists $\tau \leq \Gamma$ such that $\max\{\|x_\tau - \tilde{x}\|, \|x_\tau' - \tilde{x}\|\} \geq \mathcal{R}$.

From this, we immediately obtain a corollary that applies to any PSGD sequence:

**Corollary 1.** For any PSGD sequence $\{x_i\}$ starting at $\tilde{x}$ with $\|\nabla F(\tilde{x})\| \leq \alpha$ and $\lambda_{\min}(\nabla^2 F(\tilde{x})) \leq -\sqrt{\rho\alpha}$, with probability at least $1/8$, there exists $t \leq \Gamma$ such that $\|x_t - \tilde{x}\| \geq \mathcal{R}$.

To ensure a high-probability escape from a saddle point, we repeat $\Gamma$-`descent` for $Q$ rounds:

**Lemma 2** (Escape Amplification via Repetition). Given any $\omega_0 \in (0, 1)$, repeating $\Gamma$-`descent` independently for $Q = \frac{26}{5}\log(\frac{1}{\omega_0})$ rounds ensures escape with probability at least $1 - \omega_0$.

The proof is deferred to Appendix C.2. We now analyze the total number of PSGD steps needed for convergence. Let $\nu_t := \zeta_t + \xi_t$ denote the combined noise in the gradient estimate.

**Lemma 3** (Descent Lemma). For any $t_0$, the following holds:

$$F(x_{t_0+t}) - F(x_{t_0}) \leq -\frac{\eta}{2} \sum_{i=0}^{t-1} \|\nabla F(x_{t_0+i})\|^2 + \frac{\eta}{2} \sum_{i=1}^{t} \|\nu_{t_0+i}\|^2 \qquad (7)$$

Since $\nu_t$ can be bounded with high probability, we have:

**Corollary 2.** For any $t_0$ and some constant $c$, with probability at least $1 - 2e^{-\iota}$,

$$F(x_{t_0+t}) - F(x_{t_0}) \leq -\frac{\eta}{2} \sum_{i=0}^{t-1} \|\nabla F(x_{t_0+i})\|^2 + c\eta\psi^2(t + \iota). \qquad (8)$$

Proofs of Lemma 3 and Corollary 2 are in Appendix C.3 and C.4. These imply that large gradients lead to rapid function decrease. We next show in Lemma 4 that a successful saddle point escape via $\Gamma$-descent leads to a significant decrease in function value, whose proof is in Appendix C.5.

**Lemma 4** (Value Decrease per Escape). Let a $\Gamma$-descent starting from $x_{t_0}$ succeed after $\tau \leq \Gamma$ steps. With probability at least $1 - 2e^{-\iota}$, $F(x_{t_0+\tau}) - F(x_{t_0}) \leq -\frac{s}{8\iota^3}\sqrt{\frac{\alpha^3}{\rho}} = -\Phi$.

We bound the total number of PSGD steps required for convergence, based on the following estimate:

**Lemma 5** (Gradient Estimate Error Bound). With probability at least $1 - \omega/2$, for all $t \in [T]$, $\|\nu_t\| \leq C\sqrt{2\log\left(\frac{4T}{\omega}\right)}\psi \leq \chi$.

**Lemma 6** (Maximum Number of Descent Steps). Given failure probability $\omega$, set $Q = \frac{26}{5}\log\left(\frac{16\iota^3(F_0 - F^*)}{s\omega}\sqrt{\frac{\rho}{\chi^3}}\right)$. Gauss-PSGD returns an $\alpha$-SOSP within at most $\tilde{O}(1/\alpha^{2.5})$ PSGD steps.

Proofs of Lemmas 5 and 6 are in Appendix C.6 and C.7, respectively.

**Remark 1** (On Gradient Complexity). While Lemma 6 appears to improve gradient complexity from $O(1/\alpha^4)$ in [24] to $O(1/\alpha^{2.5})$, the two results are not directly comparable. In [24], the error target $\alpha$ is treated as an input and can be arbitrarily small, with gradient variance $\sigma$ typically treated as a constant. In contrast, in our setting, the perturbation $r$ and variance $\sigma$ are fixed by privacy constraints, and $\alpha$ emerges as a function of these. Thus, our gradient complexity fundamentally depends on $\sigma$ and $r$, though we express it in terms of $\alpha$ for clarity.

Combining all the above, we obtain the final convergence guarantee:

**Theorem 1** (Convergence Guarantee of Gauss-PSGD). Let Assumptions 1 and 2 hold. For any failure probability $\omega \in (0, 1)$, using the parameter settings in (5) and setting $Q = \frac{26}{5}\log\left(\frac{16\iota^3(F_0 - F^*)}{s\omega}\sqrt{\frac{\rho}{\chi^3}}\right)$, then with probability at least $1 - \omega$, Gauss-PSGD (Algorithm 1) returns an $\alpha$-SOSP of $F(\cdot)$, where $\alpha = 4\chi$, within at most $\tilde{O}(1/\alpha^{2.5})$ PSGD steps.

**Remark 2** (On the setting of parameters). The parameters introduced in (5) are chosen in accordance with our convergence and privacy analysis. Specifically, the escape threshold $\chi$ matches the gradient estimation error, ensuring a uniform expected decrease in the objective value per PSGD step (cf. Lemma 5 and Lemma 6). The model drift threshold $\kappa$ balances the cumulative error from the gradient oracles $\mathcal{O}_1$ and $\mathcal{O}_2$, while the maximum drift threshold $\mathcal{R}$ and maximum escape steps $\Gamma$ jointly control the curvature-dependent term $\mathcal{P}_h(t)$ and keep the stochastic gradient noise $\mathcal{P}_{sg}(t)$ bounded (see Eq. (41) and (43)). Finally, the repeat number $Q$ is chosen to grow logarithmically in the failure probability parameter to amplify the overall success probability, as established in Lemma 2.

## 5 Rectified Error Rate for finding SOSP in DP Stochastic Optimization

### 5.1 Adaptive Gradient Oracle: Ada-DP-SPIDER

In this section, we derive the upper bound on the error rate for DP stochastic optimization by instantiating the Gauss-PSGD framework with a specific gradient oracle. We adopt an adaptive version

of the DP-SPIDER estimator, referred to as `Ada-DP-SPIDER`, which is presented in Algorithm 2. This adaptive version refines the original SPIDER by dynamically adjusting gradient queries based on model drift. Unlike standard SPIDER, which queries $\mathcal{O}_1$ at fixed intervals and may suffer from growing estimation error over time, `Ada-DP-SPIDER` tracks the cumulative model drift defined as

$$\text{drift}_t := \sum_{i=\tau(t)}^{t} \|x_i - x_{i-1}\|^2, \tag{9}$$

where $\tau(t)$ is the last iteration at which the full gradient oracle $\mathcal{O}_1$ was queried.

The intuition is that, for smooth functions, the error of $\mathcal{O}_2$, which estimates $\nabla F(x_{t-1}) - \nabla F(x_{t-2})$, is proportional to $\|x_{t-1} - x_{t-2}\|$. When the model drift is small, $\mathcal{O}_2$ remains accurate, allowing for continued use to reduce variance (steps 9-11). However, when the drift becomes large, further use of $\mathcal{O}_2$ can accumulate significant errors. To mitigate this, the algorithm triggers a fresh query to $\mathcal{O}_1$ (steps 4-7). A threshold $\kappa$ is used in step 3 to determine when the drift is large. This enables adaptive switching between oracles based on the model drift, ensuring the total error remains well controlled.

Our approach differs fundamentally from that of [30]. In their method, in addition to using model drift to trigger $\mathcal{O}_1$, they also invoke $\mathcal{O}_1$ when approaching potential saddle points and inject an additional Gaussian noise on top of the DP gradient estimator to escape. To prevent excessive noise injection, they introduce a Frozen state to restrict how frequently this occurs. In contrast, our method leverages the inherent Gaussian noise from the DP gradient estimator for saddle point escape and uses model drift as the sole trigger for querying $\mathcal{O}_1$. This results in a simpler, more efficient estimator without auxiliary state tracking or redundant noise injection.

## 5.2 Error Rate Analysis for DP-SOSP with `Ada-DP-SPIDER`

To minimize the error rate $\alpha$ for DP-SOSP using `Ada-DP-SPIDER`, we must carefully tune algorithmic parameters, including the mini-batch sizes $b_1$, $b_2$, and the drift threshold $\kappa$. These parameters directly influence the gradient estimation error, which, according to Theorem 1, dominates the learning error. The following lemma characterizes how these parameters affect the estimation quality:

**Lemma 7.** Let Assumption 1 hold. For all $t \in [T]$, the gradient estimate $\hat{g}_t$ given by `Ada-DP-SPIDER` satisfies: $\sigma \leq O\left(\sqrt{\frac{G^2 \log^2 d}{b_1} + \frac{M^2 \log^2 d}{b_2}\kappa}\right), r \leq O\left(\sqrt{\frac{G^2 \log(1/\delta)}{b_1^2 \epsilon^2} + \frac{M^2 \log(1/\delta)}{b_2^2 \epsilon^2}\kappa}\right).$

The proof is given in Appendix D.1. To ensure that $b_1$ and $b_2$ remain valid mini-batch sizes under a fixed sample budget, we must control the number of times $\mathcal{O}_1$ is queried. Lemma 8 bounds the count:

**Lemma 8.** Let Assumption 1 and 2 hold. Define $\mathcal{T} := \{t \in [T] : \text{drift}_t \geq \kappa\}$ as the set of rounds where the drift exceeds the threshold $\kappa$. With high probability (as in Theorem 1), $|\mathcal{T}| \leq O\left(U\eta/\kappa\right)$.

Proof is in Appendix D.2. Guided by Lemmas 7 and 8, we now derive the error bound for $\alpha$ via appropriate choices of $b_1$, $b_2$, and $\kappa$ in Theorem 2. The proof is provided in Appendix D.3.

**Theorem 2.** Let Assumption 1 and 2 hold. Define $b_1 = \frac{n\kappa}{2U\eta}$, $b_2 = \frac{n\eta\chi^2}{2U}$ and $\kappa = \max\left\{\frac{G^{3/2}U^{1/2}\rho^{1/2}}{M^{5/2}n^{1/2}}, \frac{G^{14/15}d^{2/5}U^{4/5}\rho^{8/15}}{M^{34/15}(n\epsilon)^{4/5}}\right\}$. Then, running `Gauss-PSGD` with gradient oracle instantiated by `Ada-DP-SPIDER` ensures $(\epsilon, \delta)$-DP for constants $c_1, c_2$ and returns an $\alpha$-SOSP with $\alpha = \tilde{O}\left(\frac{1}{n^{1/3}} + \left(\frac{\sqrt{d}}{n\epsilon}\right)^{2/5}\right)$[1].

**Remark 3** (No Cyclic Dependency Among Parameters). All algorithmic parameters are consistently defined in terms of the problem parameters $n$, $d$, and $\epsilon$. Specifically, `Gauss-PSGD` parameters such as the step size $\eta$ and the noise scale $\chi$ depend on the target error $\alpha$ (see (5)), and the gradient oracle parameters $b_1$ and $b_2$ are defined through $\eta$ and $\chi$, and thus also indirectly depend on $\alpha$. In the proof of Theorem 2, by utilizing the relationship $\alpha = \tilde{O}(\sqrt{\sigma^2 + r^2 d})$, we obtain the closed-form expression of $\alpha$ that depends solely on the problem parameters $n$, $d$, and $\epsilon$. As a result, all algorithm parameters are ultimately determined by $n$, $d$, and $\epsilon$, and there is no cyclic dependency in the parameter design.

---

[1]For clarity, the bound stated here omits constant factors stemming from the Lipschitzness, smoothness, and Hessian Lipschitz assumptions. The complete expression, including these constants and their dependencies, is provided in the proof in Appendix. The same convention applies to Theorem 3.

| **Algorithm 2:** `Ada-DP-SPIDER` | **Algorithm 3:** Distributed `Ada-DP-SPIDER` |
|---|---|
| **Input:** DP budget $\epsilon$ and $\delta$, horizon $T$, model iterates $\{x_{t-1}\}_{t=1}^T$, drift threshold $\kappa$ | **Input:** DP budget $\epsilon$ and $\delta$, horizon $T$, model iterates $\{x_{t-1}\}_{t=1}^T$, drift threshold $\kappa$ |

**Algorithm 2: `Ada-DP-SPIDER`**

1 $t \leftarrow 1$, drift $\leftarrow \kappa$;
2 **while** $t \leq T$ **do**
3    **if** *drift* $\geq \kappa$ **then**
     /* Using oracle $\mathcal{O}_1$ */
4      Sample mini-batch $\mathcal{B}_t$ of size $b_1$ from $\mathcal{D}$;
5      Sample
     $\xi_t \sim \mathcal{N}(0, c_1 \frac{G^2 \log \frac{1}{\delta}}{b_1^2 \epsilon^2} \mathbf{I}_d)$;
6      $\hat{g}_t \leftarrow \mathcal{O}_1(x_{t-1}, \mathcal{B}_t) + \xi_t$;
7      drift $\leftarrow 0$;
8    **else**
     /* Using oracle $\mathcal{O}_2$ */
9      Sample mini-batch $\mathcal{B}_t$ of size $b_2$ from $\mathcal{D}$;
10      Sample $\xi_t \sim \mathcal{N}(0, c_2 \frac{M^2 \log \frac{1}{\delta}}{b_2^2 \epsilon^2} \|x_{t-1} - x_{t-2}\|^2 \mathbf{I}_d)$;
11      $\hat{g}_t \leftarrow \hat{g}_{t-1} + \mathcal{O}_2(x_{t-1}, x_{t-2}, \mathcal{B}_t) + \xi_t$;
12    drift $\leftarrow$ drift $+ \eta^2 \|\hat{g}_t\|^2$;
13    $t \leftarrow t + 1$;
**Output:** $\hat{g}_1, \hat{g}_2, \cdots, \hat{g}_T$

**Algorithm 3: Distributed `Ada-DP-SPIDER`**

1 $t \leftarrow 1$, drift $\leftarrow \kappa$;
2 **while** $t \leq T$ **do**
3    **if** *drift* $\geq \kappa$ **then**
4      **for** every *client* $j$ **in parallel do**
5       Sample mini-batch $\mathcal{B}_{j,t}$ of size $b_1$ from $\mathcal{D}_j$;
6       Sample $\xi_{j,t} \sim \mathcal{N}(0, c_1 \frac{G^2 \log \frac{1}{\delta}}{b_1^2 \epsilon^2} \mathbf{I}_d)$;
7       $\hat{g}_{j,t} \leftarrow \mathcal{O}_1(x_{t-1}, \mathcal{B}_{j,t}) + \xi_{j,t}$;
8       Send $\hat{g}_{j,t}$ to the server;
9      drift $\leftarrow 0$;
10    **else**
11      **for** every *client* $i$ **in parallel do**
12       Sample mini-batch $\mathcal{B}_{j,t}$ of size $b_2$ from $\mathcal{D}_j$;
13       Sample $\hat{\xi}_{j,t} \sim \mathcal{N}(0, c_2 \frac{M^2 \log \frac{1}{\delta}}{b_2^2 \epsilon^2} \|x_{t-1} - x_{t-2}\|^2 \mathbf{I}_d)$;
14       $\hat{g}_{j,t} \leftarrow \hat{g}_{j,t-1} + \mathcal{O}_2(x_{t-1}, x_{t-2}, \mathcal{B}_{j,t}) + \xi_{j,t}$;
15       Send $\hat{g}_{j,t}$ to the server;
16    $\hat{g}_t \leftarrow \frac{1}{m} \sum_{j=1}^m \hat{g}_{j,t}$;
17    drift $\leftarrow$ drift $+ \eta^2 \|\hat{g}_t\|^2$;
18    $t \leftarrow t + 1$;
**Output:** $\hat{g}_1, \hat{g}_2, \cdots, \hat{g}_T$

## 6 Extension to Distributed SGD

By adapting the centralized gradient oracle `Ada-DP-SPIDER` (Algorithm 2) to the distributed setting, we obtain `Distributed Ada-DP-SPIDER` (Algorithm 3), enabling our `Gauss-PSGD` framework to extend seamlessly to distributed learning scenarios. The primary difference lies in the computation and communication scheme: in the distributed variant, each client performs local gradient estimation with private noise and communicates the privatized estimate to the server, which then aggregates the results. This avoids centralized access to raw data while still leveraging collective information.

The learning algorithm using `Distributed Ada-DP-SPIDER` can be viewed as an adaptive extension of the DIFF2 algorithm [37], which uses standard SPIDER and is limited to convergence to DP-FOSP under *homogeneous* data. To the best of our knowledge, our method is the first to achieve convergence to a DP-SOSP in a distributed setting with *heterogeneous* data.

Following the same analytical strategy as in Section 5, we first quantify in Lemma 9 the gradient estimation quality in the distributed case. The proof is provided in Appendix E.1.

**Lemma 9.** Let Assumption 1 hold. For all $t \in [T]$, the distributed `Ada-DP-SPIDER` ensures that the gradient estimate $\hat{g}_t$ satisfies $\sigma \leq O\left(\sqrt{\frac{G^2 \log^2 d}{m \cdot b_1} + \frac{M^2 \log^2 d}{m \cdot b_2} \kappa}\right), r \leq O\left(\sqrt{\frac{G^2 \log \frac{1}{\delta}}{m \cdot b_1^2 \epsilon^2} + \frac{M^2 \log \frac{1}{\delta}}{m \cdot b_2^2 \epsilon^2} \kappa}\right)$.

Based on this, we derive the error bound for $\alpha$ in the distributed setting. The proof is in Appendix E.2.

**Theorem 3.** Let Assumption 1 and 2 hold. Define $b_1 = \frac{n\kappa}{2U\eta}$, $b_2 = \frac{n\eta\chi^2}{2U}$ and $\kappa = \max\left\{\frac{G^{3/2} U^{1/2} \rho^{1/2}}{M^{5/2}(mn)^{1/2}}, \frac{G^{14/15} d^{2/5} U^{4/5} \rho^{8/15}}{M^{34/15}(\sqrt{m}n\epsilon)^{4/5}}\right\}$. Then, running `Gauss-PSGD` with gradient oracle instantiated by distributed `Ada-DP-SPIDER` ensures $(\epsilon, \delta)$-ICRL-DP for some constants $c_1, c_2$, and returns an $\alpha$-SOSP with $\alpha = \tilde{O}\left(\frac{1}{(mn)^{1/3}} + \left(\frac{\sqrt{d}}{\sqrt{m}n\epsilon}\right)^{2/5}\right)$.

---

**Algorithm 4:** Private Model Selection in Distributed Learning

---

**Input:** Model iterates $\{x_t\}_{t=1}^T$, DP budget $\epsilon, \delta$

1   **for** $t \leftarrow 1, \cdots, T$ **do**

2     **for every** *client* $j$ **in parallel do**

3       Compute $\nabla \bar{F}_j(x_t) \leftarrow \nabla \hat{f}_{S_j}(x_t) + \theta_{i,t}$, where $\theta_{i,t} \sim \mathcal{N}\left(0, c_1 \frac{G^2 T \log(1/\delta)}{n^2 \epsilon^2} \mathbf{I}_d\right)$ ;

4       Compute $\nabla^2 \bar{F}_j(x_t) \leftarrow \nabla^2 \hat{f}_{S_j}(x_t) + \mathbf{H}_{j,t}$, where $\mathbf{H}_{j,t}$ is a symmetric matrix with its upper triangle (including the diagonal) being i.i.d. samples from $\mathcal{N}\left(0, c_2 \frac{M^2 dT \log(1/\delta)}{n^2 \epsilon^2}\right)$ and each lower triangle entry is copied from its upper triangle counterpart;

5       Send $\nabla \bar{F}_j(x_t)$ and $\nabla^2 \bar{F}_j(x_t)$ to the server;

6     $\nabla \bar{F}(x_t) \leftarrow \frac{1}{m} \sum_{j=1}^m \nabla \bar{F}_j(x_t), \nabla^2 \bar{F}(x_t) \leftarrow \frac{1}{m} \sum_{j=1}^m \nabla^2 \bar{F}_j(x_t)$;

7     **if** $\|\nabla \bar{F}(x_t)\|_2 \leq \alpha + \frac{G \log(8d/\omega')}{\sqrt{mn}} + \frac{G\sqrt{dT \log(1/\delta) \log(16/\omega')}}{\sqrt{mn}\epsilon}$ **and**

      $\lambda_{\min}\left(\nabla^2 \bar{F}(x_t)\right) \geq -\left(\sqrt{\rho\alpha} + M\sqrt{\frac{\log(8d/\omega')}{mn}} + \frac{Md\sqrt{T \log(1/\delta) \log(32/\omega')}}{\sqrt{mn}\epsilon}\right)$ **then**

8       **Return** $x_t$

---

**Remark 4.** The error rate shown in Theorem 3 highlights the collaborative synergy among clients, indicating the learning performance benefits from distributed learning. Specifically, the first non-private term of $\alpha$ exhibits a linear dependence on $m$ before $n$, while the second term, which accounts for the privacy cost, demonstrates a square root dependence $\sqrt{m}$ before $n$. This separation reflects the impact of data heterogeneity in distributed setting. The benefit of distributed collaboration under DP constraints is consistent with prior results in heterogeneous federated learning [16].

We conclude by demonstrating the advantages of our `Gauss-PSGD` framework in distributed learning by eliminating the need for a separate private model selection procedure. Without the guarantee of directly outputting an $\alpha$-SOSP, one must resort to evaluating all model iterates generated during the learning process and privately selecting an approximate SOSP from them. As discussed in Appendix A, the AboveThreshold mechanism used in [30] for the single-machine case is not applicable in distributed settings due to decentralized data access. To overcome this, we adapt [46, Algorithm 5] to the distributed setting, resulting in Algorithm 4. In this scheme, each client computes privatized gradients and Hessian estimates using additional local data, which are then aggregated by the server to evaluate the stationary point conditions. Suppose a distributed learning algorithm produces a sequence $\{x_t\}_{t \in [T]}$ that contains at least one $\alpha$-DP-SOSP. The following result characterizes the quality of the point selected by Algorithm 4, whose proof is provided in Appendix E.3:

**Theorem 4.** Algorithm 4 satisfies $(\epsilon, \delta)$-ICRL-DP. Let Assumption 1 hold and $mn \geq \frac{4}{9} \log \frac{8d}{\omega'}$, then with probability at least $1 - \omega'$, if there exists an $\alpha$-SOSP $x_p \in \{x_t\}_{t=1}^T$, then the selected point $x_o$ is an $\alpha'$-SOSP with $\alpha' = \tilde{O}\left(\alpha + \frac{1}{mn} + \frac{1}{\sqrt{mn}} + \frac{\alpha}{\sqrt{mn}} + \frac{\sqrt{d}}{\sqrt{mn}\epsilon\alpha^{5/4}} + \frac{d}{\sqrt{mn}\epsilon\alpha^{3/4}} + \frac{d^2}{mn^2\epsilon^2\alpha^{5/2}}\right)$.

**Remark 5.** To ensure that the selected model's error $\alpha'$ does not exceed the training error $\alpha$, the following must hold: $\frac{\sqrt{d}}{\sqrt{mn}\epsilon\alpha^{5/4}} + \frac{d}{\sqrt{mn}\epsilon\alpha^{3/4}} + \frac{d^2}{mn^2\epsilon^2\alpha^{5/2}} \leq \tilde{O}(\alpha)$. This implies a constraint on the model dimension: $d \leq \min\{(\sqrt{mn}\epsilon)^2, (\sqrt{mn}\epsilon)^{6/13}\}$. Thus, in high-dimensional regimes, private model selection degrades the overall error rate, marking the limitation of selection-based approaches.

**Remark 6.** The error bound $\alpha'$ in Theorem 4 can be improved by estimating the smallest eigenvalue of the Hessian via Hessian-vector products using iterative methods such as the power method [26]. This reduces the dimensional dependence in the noise scale from $O(d)$ to $O(\sqrt{d})$. However, the remaining $\sqrt{d}$ factor is sill problematic in high-dimensional settings. In contrast, in the single-machine case, private model selection only requires perturbing scalar quantities, making the error independent of dimension, preserving the error guarantee of the learning algorithm. In distributed settings, sharing perturbed vectors becomes unavoidable. This emphasizes the necessity and superiority of our `Gauss-PSGD` framework that inherently avoids the need for any separate model selection step.

## Acknowledgments and Disclosure of Funding

Youming Tao was supported in part by the National Science Foundation of China (NSFC) under Grant 623B2068. Dongxiao Yu is supported in part by the Major Basic Research Program of Shandong Provincial Natural Science Foundation under Grant ZR2025ZD18. Xiuzhen Cheng is supported in part by the Major Basic Research Projects of Shandong Natural Science Foundation under Grant ZR2022ZD02. Di Wang is supported in part by the funding BAS/1/1689-01-01 and funding from KAUST - Center of Excellence for Generative AI, under award number 5940.

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
