# OpenReview forum: "Second-Order Convergence in Private Stochastic Non-Convex Optimization"
_NeurIPS.cc/2025/Conference — NeurIPS 2025 poster_

### Official Review · Reviewer_jvqK · 2025-06-30

**Clarity:** 2
**Significance:** 2
**Originality:** 3
**Rating:** 4
**Confidence:** 3

**Summary:**

This paper proposes a generic perturbed stochastic gradient descent (PSGD) framework based on Gaussian noise injection and general gradient oracles to address the problem of second-order stationary points (SOSP) in differential privacy (DP) stochastic non-convex optimization, addressing two limitations of existing methods: （1）Inaccuracy of convergence error rates due to neglect of gradient variance; （2）Utility degradation caused by reliance on auxiliary private model selection procedures. The paper also discusses the limitation of private model selection in high-dimensional distributed settings, and provides numerical experiments to support its approach.

**Questions:**

1. How are the hyperparameters, including the escape threshold, model drift threshold, maximum drift threshold, maximum escape steps, and maximum repeat number of escape, determined? .
2. Beyond what is covered in the appendix, what important usage scenarios or risks where model drift-based DP-SOSP is especially helpful or potentially problematic?

**Ethical Concerns:**

["NO or VERY MINOR ethics concerns only"]

**Final Justification:**

The rebuttal satisfactorily addresses all of my concerns, and I have increased my score by one point accordingly.

**Limitations:**

yes

**Quality:**

3

**Strengths And Weaknesses:**

**Strengths**

1. The paper demonstrates specific flaws in their saddle point escape analysis in prior works. Based on these findings, the paper motivates and delivers an algorithmic redesign that resolves these inherent limitations.
2. The paper introduces of Gauss-PSGD , which uses model drift distance as the escape trigger and allows the algorithm to avoid the need for high-overhead, additional DP model selection.
3. The paper is technically sound and clear in its theorems, algorithmic procedures, and justification for each design and proof step.
4. Another significant contribution is the extension to federated/distributed learning with arbitrary data heterogeneity.



**Weakness**

1. Parameter setting explanations (e.g., for step sizes, batch sizes, drift thresholds in Section 3.3 and 4.2) are quite technical and packed with dense notation. More intuitive explanations are required to help understand trade-offs.
2. Although the results advance DP-SOSP theory, the discussion in the main paper is focused on theoretical mechanisms and error rates, with little commentary on how these results could impact industrial-scale machine learning tasks where privacy and distributed non-convex optimization interact.
3. There is only a brief mention of numerical experiments in the abstract and summary, and detailed results are placed in the appendix. It is better to remove them to the main submission, especially the comparison with [29].
4. Several important pieces, such as the detailed discussion of prior works, and experimental evaluations, are delegated to the appendix. A more self-contained and empirically substantiated presentation would considerably strengthen the case for publication.

---

> ### Author Rebuttal · Authors · 2025-07-29
>
> We thank Reviewer jvqK for the detailed and constructive review. We are grateful for the recognition of our contributions, including the correction of flaws in prior saddle point escape analysis, the design of Gauss-PSGD based on model drift, the elimination of high-overhead private model selection, and the extension to distributed learning with heterogeneous data. We also appreciate the acknowledgment of the clarity and rigor of our theorems and algorithmic design.
>
> ## Response to Weaknesses
>
> **1. Regarding dense parameter settings in Section 3.3 and 4.2：** Thank you for the suggestion. In the final version, we will supplement the main text with more intuitive explanations of the parameters, including how they relate to algorithmic behavior and theoretical guarantees, to improve accessibility for broader audiences.
>
> **2. Regarding practical impact for industrial-scale ML tasks:** We agree that connecting our theoretical results to real-world applications is important. This work focuses primarily on improving and correcting the theoretical foundations of DP-SOSP, particularly addressing limitations in [29] and providing a refined understanding of saddle point escape behavior. That said, we will add a brief discussion to highlight potential industrial use cases, such as privacy-preserving distributed training, and we plan to explore more empirical evaluations in future work.
>
> **3 & 4. Regarding the organization of the paper:** We appreciate the helpful suggestions. Due to strict page constraints, we initially placed some experimental results and prior work discussions in the appendix. We agree that summarizing key empirical findings, especially the comparison with [29], in the main paper would strengthen the presentation. If accepted, we will use the additional content page to reincorporate the most essential experiments and discussion of prior works into the main submission, making it more self-contained and empirically grounded.
>
> ## Response to Questions
>
> **Q1:** The parameter settings are derived from our convergence and privacy analysis:
> - The escape threshold $\chi$ is set to match the gradient estimation error, as established in Lemma 5, which ensures a uniform average decrease in function value of at least $O(\chi^2\eta)$ per PSGD step, regardless of whether the algorithm is in an escape phase, as shown in Lemma 6.
> - The model drift threshold $\kappa$ is chosen to balance the cumulative error introduced by the gradient oracle $\mathcal{O}_1$ and $\mathcal{O}_2$, as shown around lines 790 and 820.
> - The maximum drift threshold $\mathcal{R}$ and the maximum escape steps $\Gamma$ are designed to (1) control the curvature-dependent term $\mathscr{P}_h(t)$, as in Eq. (41); and (2) ensure that the stochastic gradient noise term $\mathscr{P} _{sg}(t)$ remains bounded, as in Eq. (43).
> - The maximum repeat number $Q$ is selected logarithmically in failure probability to amplify success probability, as specified in Lemma 2.
>
> **Q2:** The model drift criterion is particularly useful in settings where global function values are inaccessible, but model updates are observable, such as the population risk minimization setting considered in our paper. One potential issue is that small, flat local minima may trigger unnecessary escapes. However, as noted in our response to Reviewer jpLh's Q1, this does not affect our convergence guarantee to a DP-SOSP. Moreover, this property could potentially be leveraged to encourage flatter minima, in line with recent work on sharpness-aware minimization for improved generalization. We view this as a promising direction for future research.

---

> > ### Comment · Reviewer_jvqK · 2025-08-05
> >
> > I acknowledge the theoretical contributions of this paper. However, from a machine learning practitioner's perspective, it remains unclear how the theoretical analysis or insights can be applied in practice. This aligns with the concern raised in Weakness 2 by Reviewer Rud7. In the rebuttal, the authors mention that they will include a discussion in the final version, but no detailed analysis was provided. Therefore, I have decided to maintain my original score.

---

> > > ### Author Response · Authors · 2025-08-05
> > >
> > > We thank the reviewer for the follow-up. As requested, we provide a more detailed discussion on the practical relevance of our  algorithm and theoretical results, particularly in the context of industrial-scale distributed learning systems.
> > >
> > > Modern industrial machine learning systems, such as those used in applications like Google Gboard and Apple Siri, often involve highly non-convex objectives (e.g., deep neural networks), distributed computation (e.g., federated learning), and strict user-level privacy guarantees (e.g., differential privacy). These systems must operate over massively heterogeneous and decentralized data while maintaining scalability and accuracy under privacy constraints.
> > >
> > > Our theoretical results address these challenges by establishing, for the first time, that the Gaussian-perturbed DP-SGD algorithm can provably escape saddle points and converge to second-order stationary points (DP-SOSP), even in distributed and heterogeneous settings. In contrast, prior work [29] focuses on a centralized setting and requires additional noise beyond that used for DP guarantees. Their added noise not only complicates real-world implementation but also degrades model accuracy due to unnecessary perturbations. Moreover, the centralized nature of [29] severely limits scalability, which is a critical bottleneck in industrial-scale applications.
> > >
> > > By contrast, our analysis applies directly to standard DP-SGD in distributed environments with arbitrary heterogeneous data, without requiring extra noise or architectural changes. This preserves both model utility and training scalability, offering practical benefits in large-scale deployment scenarios.
> > >
> > > We hope this addresses your concerns, and we sincerely thank you for your time and engagement.

---

> > > > ### Comment · Reviewer_jvqK · 2025-08-05
> > > >
> > > > Thank you for the clarification. While many existing works apply differential privacy to federated learning, the primary novelty of this paper appears to lie in its theoretical analysis concerning convergence to second-order stationary points (DP-SOSP). However, I am still unclear on how this insight directly benefits machine learning practitioners. For instance, does DP-SOSP imply improved convergence speed or better empirical performance? In other words, how can this theoretical guarantee be leveraged to improve practical systems?
> > > >
> > > > Additionally, the paper claims that the proposed method "preserves both model utility and training scalability," but I was unable to find experimental results that clearly support this statement.

---

> > > > > ### Author Response · Authors · 2025-08-05
> > > > >
> > > > > We sincerely thank the reviewer for the prompt follow-up and the opportunity to further clarify the practical relevance of our results.
> > > > >
> > > > > (1) On the practical benefits of DP-SOSP:
> > > > >
> > > > > **Saddle points and local maxima can lead to highly suboptimal solutions in many non-convex optimization problems** [JJKN15; SQW16a]. [DPGC14] further argue that saddle points are ubiquitous in high-dimensional, non-convex landscapes and constitute a major bottleneck in training deep neural networks. Our DP-SOSP guarantee ensures that the algorithm escapes saddle points and converges to an (approximate) local minimum, which is thus crucial for reliable optimization in practice.
> > > > >
> > > > > This guarantee is particularly meaningful given that, although non-convex optimization is NP-hard in general, **many real-world problems only require convergence to a local minimum**. Prior work has shown that, in various applications, including tensor decomposition [GHJY15], dictionary learning [SQW16b], phase retrieval [SQW16a], matrix sensing [BNS16; PKCS17], matrix completion [GLM16], and certain deep networks [K16], all local minima are in fact global minima. Furthermore, [CHMA15] suggest that even in more general deep networks, most local minima have objective values comparable to the global minimum.
> > > > >
> > > > > Taken together, these results imply that **convergence to a second-order stationary point (rather than a first-order one) often leads to better practical outcomes**. Regarding convergence speed, while our work focuses on convergence **error** under differential privacy rather than iteration complexity, we note that in the non-private setting, convergence to a SOSP can typically be achieved with similar complexity to FOSP (up to polylogarithmic factors); see reference [23] in our paper for details.
> > > > >
> > > > > > [JJKN15] Jain, P., Jin, C., Kakade, S. M., & Netrapalli, P. (2015). Computing matrix squareroot via non convex local search. arXiv preprint arXiv:1507.05854.
> > > > >
> > > > > > [SQW16a] Sun, J., Qu, Q., & Wright, J. (2016, July). A geometric analysis of phase retrieval. In 2016 IEEE International Symposium on Information Theory (ISIT) (pp. 2379-2383). IEEE.
> > > > >
> > > > > > [DPGC14] Dauphin, Y. N., Pascanu, R., Gulcehre, C., Cho, K., Ganguli, S., & Bengio, Y. (2014). Identifying and attacking the saddle point problem in high-dimensional non-convex optimization. Advances in neural information processing systems, 27.
> > > > >
> > > > > > [GHJY15] Ge, R., Huang, F., Jin, C., & Yuan, Y. (2015, June). Escaping from saddle points—online stochastic gradient for tensor decomposition. In Conference on learning theory (pp. 797-842). PMLR.
> > > > >
> > > > > > [SQW16b] Sun, J., Qu, Q., & Wright, J. (2016). Complete dictionary recovery over the sphere I: Overview and the geometric picture. IEEE Transactions on Information Theory, 63(2), 853-884.
> > > > >
> > > > > > [BNS16] Bhojanapalli, S., Neyshabur, B., & Srebro, N. (2016). Global optimality of local search for low rank matrix recovery. Advances in Neural Information Processing Systems, 29.
> > > > >
> > > > > > [PKCS17] Park, D., Kyrillidis, A., Carmanis, C., & Sanghavi, S. (2017, April). Non-square matrix sensing without spurious local minima via the Burer-Monteiro approach. In Artificial Intelligence and Statistics (pp. 65-74). PMLR.
> > > > >
> > > > > > [GLM16] Ge, R., Lee, J. D., & Ma, T. (2016). Matrix completion has no spurious local minimum. Advances in neural information processing systems, 29.
> > > > >
> > > > > > [K16] Kawaguchi, K. (2016). Deep learning without poor local minima. Advances in neural information processing systems, 29.
> > > > >
> > > > > > [CHMA15] Choromanska, A., Henaff, M., Mathieu, M., Arous, G. B., & LeCun, Y. (2015, February). The loss surfaces of multilayer networks. In Artificial intelligence and statistics (pp. 192-204). PMLR.
> > > > >
> > > > > (2) On experimental support for "preserving model utility and scalability":
> > > > >
> > > > > Although our primary contribution is theoretical, our experiments also provide supporting evidence for both utility and scalability. The baseline method from [29] can only be implemented in the single-client setting (m=1), as it relies on a centralized framework. In contrast, our algorithm is implemented and tested across various client settings (m=1,2,5,10).
> > > > >
> > > > > In terms of **model utility**, our method achieves **superior performance** compared to the baseline in the m=1 case, as evidenced by both test accuracy and loss. Regarding **scalability**, the performance of our algorithm improves as the number of clients increases, demonstrating its ability to scale effectively to distributed learning scenarios while maintaining high utility.
> > > > >
> > > > > We will clarify these empirical observations in the revised version and appreciate the reviewer’s helpful comments.

---

> > > > > > ### Comment · Reviewer_jvqK · 2025-08-06
> > > > > >
> > > > > > Thank you for your response. It is a well-accepted understanding that many real-world problems only require convergence to a local minimum. As noted by Reviewer jpLH, your paper's motivation centers on the "theoretical" appeal of second-order stationary points. However, this does not necessarily imply that achieving second-order stationarity leads to better "practical" outcomes.
> > > > > >
> > > > > > While your comparison with plain DP-SGD is appreciated, one strength of your work is its extension of prior centralized results (e.g., [29]) to the distributed multi-client setting. However, several existing works have already proposed differentially private methods for federated learning. The paper would benefit from a direct comparison in convergence error to such methods to more convincingly demonstrate the practical effectiveness of your approach.

---

> > > > > > > ### Author Response · Authors · 2025-08-06
> > > > > > >
> > > > > > > We thank the reviewer for the helpful comments and would like to provide a further clarification.
> > > > > > >
> > > > > > > To address the concern regarding empirical comparison with existing differentially private federated learning algorithms, we have added experiments comparing our method to DIFF2 [36], a recent state-of-the-art DP-FL algorithm that incorporates the original SPIDER technique for variance reduction. We evaluated both methods on the CIFAR-10 dataset using a ResNet-18 model with $m=10$ clients, under a privacy budget of $\epsilon=2$, $\delta=10^{-5}$. We report the average test accuracy over 5 independent runs and include the standard deviation to assess the stability of the results. The outcomes are summarized as follows:
> > > > > > >
> > > > > > > |   Method   |       |   Avg. Test Accuracy (%)   |       |   Std. Dev. (%)   |
> > > > > > > |:----------:|:-----:|:--------------------------:|:-----:|:-----------------:|
> > > > > > > |  DIFF2    |       |           77.1             |       |        3.6        |
> > > > > > > |   Ours     |       |           **78.9**             |       |      **1.1**     |
> > > > > > >
> > > > > > > **Our method not only achieves higher average accuracy, but also exhibits substantially lower variance across runs, indicating improved stability, which we attribute to its ability to escape saddle points.** Due to NeurIPS rebuttal constraints, we are unable to include figures or external URLs at this stage. However, complete plots will be provided in the final version of the paper.
> > > > > > >
> > > > > > > That said, we emphasize that the primary contribution of our work remains theoretical. Our goal is to advance the theoretical understanding of differentially private non-convex optimization by establishing rectified and improved analytical bounds for DP-SOSP, with a provably efficient algorithm that extends naturally to distributed settings. Our focus is on clean and general results under standard assumptions. As is the case with many NeurIPS papers, we view the empirical evaluation as a complement to our theoretical findings, not the core of our contribution. We hope the paper will be assessed based on its intended scope and theoretical significance, and that the absence of broader empirical comparisons will not be viewed as a fatal weakness.
> > > > > > >
> > > > > > > We sincerely thank the reviewer for their time and effort.

---

> ### Comment · Reviewer_jvqK · 2025-08-07
>
> Thanks to the authors for their detailed response. I acknowledge the theoretical contributions of the paper. However, as with many differentially private (DP) works that include dense mathematical derivations and theorems, the true impact of such contributions can be difficult to assess if the results do not provide clear insights into practical scenarios. In that sense, while the theory appears sound, it is challenging to evaluate its real-world significance, as many DP papers appear equally rigorous on the surface.
>
> That said, I appreciate the additional experiments provided in the rebuttal. However, another concern arises: when comparing results across DP-SGD, DIFF2, and the proposed method using ResNet-18, it appears that the proposed method outperforms the others in federated settings. This outcome contradicts my general understanding—federated learning typically yields comparable or slightly worse performance than centralized settings due to issues such as data heterogeneity and limited communication. Clarification on this point would be helpful.

---

> > ### Author Response · Authors · 2025-08-07
> >
> > We thank the reviewer for the follow-up.
> >
> > The improved performance in the federated setting is not contradictory, but rather reflects the benefit of involving more data and distributed computation in a privacy-preserving way, in line with our theoretical guarantees.
> >
> > Our theoretical convergence error bound is given by:
> >
> > $\alpha=\tilde{O}\left(\frac{1}{(mn)^{1/3}}+\left(\frac{\sqrt{d}}{\sqrt{m}n\epsilon}\right)^{2/5}\right)$
> >
> > where $m$ is the number of clients, $n$ is the number of data samples **per client**, $d$ is the dimensionality, and $\epsilon$ is the privacy budget. This result shows that increasing the number of clients $m$ leads to a smaller overall error bound, as both terms improve with larger $m$.
> >
> > In the results we reported to Reviewer jpLH, both DP-SGD and our method were evaluated in the centralized setting $m=1$. In contrast, the results reported to you compare DIFF2 and our method in the federated setting $m=10$. The improved performance of our method in the federated case is therefore expected and aligns with the theory. Additionally, the superior performance of our method compared to both DP-SGD and DIFF2 is largely attributed to its ability to escape saddle points, leading to more accurate and stable results.
> >
> > We hope this fully resolves your concern and appreciate your continued engagement.

---

> > > ### Comment · Reviewer_jvqK · 2025-08-08
> > >
> > > Thank you to the authors for their detailed response. It satisfactorily addresses all of my concerns, and I have increased my score by one point accordingly.

---

### Official Review · Reviewer_Rud7 · 2025-06-30

**Clarity:** 3
**Significance:** 2
**Originality:** 2
**Rating:** 4
**Confidence:** 2

**Summary:**

This manuscript aims to find second-order stationary points in differentially private non-convex optimization. They propose a generic non-convex stochastic optimization framework PSGD, and provide an improved error rate bound. Besides, they extend their theoretical analysis to distributed learning and obtain the first error bound for DP-SOSP under arbitrarily heterogeneous data.

**Questions:**

- This manuscript mainly compares their result with "Private (stochastic) non-convex optimization revisited: Second-order stationary points and excess risks" regarding the population risk. What about the empirical risk, compared with SOTA and Lower bound?

**Ethical Concerns:**

["NO or VERY MINOR ethics concerns only"]

**Final Justification:**

The authors have well answered my questions. I perceive that more introductions and insights related to the machine learning community are needed for a better understanding of the contribution. Overall, I will maintain my scores.

**Limitations:**

Yes.(Section I in Appendix)

**Paper Formatting Concerns:**

No.

**Quality:**

3

**Strengths And Weaknesses:**

**Strengths**
- This manuscript is well-organized, clearly presenting their theoretical analysis.

- They obtain the recified error rate compared to prior work for finding SOSP and obtain the first error bound for DP-SOSP under arbitrarily heterogeneous data.

**Weaknesses**

- I notice that authors have conducted experiments for varying privacy budget and the number of clients in the Appendix. I suggest authors briefly mention their empirical results, which validate the effectiveness of their method, in the main text for better readability.

- I acknowledge the statistical contribution of this manuscript. It would be better for machine learning community to understand if more explanations or insights are included. For example, why studying DP-SOSP is important; what insights we can get from all those theorems.

---

> ### Author Rebuttal · Authors · 2025-07-29
>
> We thank Reviewer Rud7 for the thoughtful feedback and for acknowledging the clarity of our theoretical analysis and the significance of our results, particularly the rectified error rate and the first error bound for DP-SOSP under arbitrarily heterogeneous data.
>
> ## Response to Weaknesses
>
> **1. Regarding empirical results only mentioned in the appendix:** We appreciate the reviewer’s suggestion. We will summarize the key empirical results in the main text and, if accepted, use the extra page to incorporate them more clearly.
>
> **2. Regarding need for more insights into the importance of DP-SOSP and the theorems:** Thank you for pointing this out. We agree that providing more intuitive explanations about the motivation and implications of DP-SOSP would benefit a broader audience. In the final version, we will update the introduction to emphasize why DP-SOSP is important: while first-order convergence (FOSP) may still return saddle points or local maxima, second-order guarantees are essential for ensuring meaningful local optimality in non-convex problems. We will also include a brief comment on what insights the theorems provide.
>
> ## Response to Question
>
> Our work focuses on population risk minimization because the issues we identified in the prior method and its analysis primarily arise in this setting. For ERM, we note that our Gauss-PSGD framework is also compatible with ERM analysis, and whether it can improve the bound for ERM is an interesting problem to investigate for future work. Regarding lower bound, the results presented in [29] are directly from [1], which primarily addresses convex loss functions and first-order stationary points. In contrast, establishing lower bounds for finding DP-SOSP in non-convex optimization is inherently more challenging and remains an open problem, which we leave for future work.

---

> > ### Comment · Reviewer_Rud7 · 2025-08-08
> >
> > Thank the authors for the response. My concerns have been well explained and I will keep my positive scores.

---

### Official Review · Reviewer_DNHR · 2025-07-01

**Clarity:** 3
**Significance:** 3
**Originality:** 3
**Rating:** 5
**Confidence:** 3

**Summary:**

This paper studies the second order convergence guarantees of a differentially private stochastic non-convex optimization problem. Authors assume a sufficiently smooth loss function (Lipschitz, smooth and Lipschitz Hessian), and propose a generic optimization algorithm Gauss-PSGD that obtains the gradients from arbitrary gradient oracles. The algorithm tries to detect if the optimization has ran into a saddle point, and tries to escape it by continuing the stochastic optimization and checking if the parameter value has moved enough from the supposed saddle point. Authors prove several theoretical properties for the Gauss-PSGD: high-probability bound of escaping the saddle point, high-probability convergence guarantee into a 2nd order optimal point, and using specific DP gradient oracle show an improved error rate for DP non-convex optimization. Authors also extend the setting for a distributed learning, where the data can be heterogeneous among the clients. In this setting authors can recover very similar error rate as with the general DP non-convex optimization.

**Questions:**

1. Is there a reason to have the constants $c_1$ and $c_2$ in Thm 2.? I guess the constants arise from the moments-accountant-like analysis, and hence for a fixed $\epsilon$ and $\delta$ mainly affect the noise level, which (I guess) is not explicitly present in Thm 2. Maybe there is some reason to have them there, but to me it seemed just confusing to have the explicitly stated when the expression does not depend on them.
2. The constants $c_1$ and $c_2$ in Thm. 2 remain a bit unclear to me. First, as they are not explicitly part of the bound, is is necessary to have them? Second, are these constants effectively the same as in moments accountant privacy analysis by Abadi et al. 2016?
3. It seems that the Gauss-PSGD achieves better test accuracy with $\epsilon=0.5$ than $\epsilon=2$ (Fig. 2, m=10 result). Do you have any idea why this is?
4. Does the single hidden layer NN you have selected for the experiments actually satisfy some of the regularity conditions (Assumptions 1 and 2)? If the objective is not Lipschitz, I guess you have used the biased clipped gradient oracle? And if so, can you report the clipping threshold for clarity sake.
5. typo: "requirment"

**Ethical Concerns:**

["NO or VERY MINOR ethics concerns only"]

**Final Justification:**

Authors rebuttal addressed my concerns, and I will retain my original positive score on the paper. In my opinion this is a solid theory paper that makes a novel and significant contribution to the field. However, empirical comparisons against e.g. DP-SGD would further strengthen the paper and demonstrate if the second order guarantees lead to practically significant improvements.

**Limitations:**

Yes, although the discussion is pushed mainly to the Appendix.

**Quality:**

3

**Strengths And Weaknesses:**

## Strengths
1. The paper presents a novel 2nd order error analysis for DP non-convex optimization problem. The proposed error rates improve (although somewhat moderately) over the existing bounds, allowing to better understand the limits of DP non-convex optimization.  (**Originality + Significance**)
2. The extension to distributed setting is interesting, especially given that the prior work only has convergence results for the homogeneous data. (**Significance**)
3. The proposed optimization framework Gauss-PSGD gives a simple unified framework for the non-convex optimization. For DP (and distributed DP) authors propose a specific gradient oracle Ada-DP-SPIDER, that tries to reduce the variance in the optimization. Authors demonstrate through some simple experiments that the proposed method out-performs an existing DP non-convex optimization method targeted for 2nd order guarantees. (**Significance + Quality**)
4. Paper is rather busy, but the key points are explained and argued extremely well. (**Clarity**)

## Weaknesses
1. While the paper is mainly theoretical, it does propose an algorithm. So far, the experiments have been limited to somewhat simple models, so some more demanding tasks would strengthen the claims. Furthermore, having other baselines would allow putting the results in a better perspective. E.g. how would plain DP-SGD perform in the chosen tasks. (**Significance**)
2. I think having the conclusions (amongst many other important sections) in the Appendix is a bit weird. I know that you operate under a strict page limit, but compressing the main paper does not seem impossible, especially as the conclusions and limitations allow contain some important discussion about the proposed method. (**Clarity**)

---

> ### Author Rebuttal · Authors · 2025-07-29
>
> We thank Reviewer DNHR for the positive assessment and for highlighting the originality, significance, and clarity of our work. We are particularly grateful for the recognition of our new second-order error analysis for DP non-convex optimization, the extension to distributed heterogeneous data, the design of the unified Gauss-PSGD framework with Ada-DP-SPIDER, and the clarity of our theoretical exposition despite the technical density.
>
> ## Response to Weaknesses
>
> **1. Regarding the scope of experiments:** We agree that more demanding tasks and broader baselines would strengthen the empirical perspective. However, as the reviewer noted, our paper is primarily theoretical. Our experiments are designed to validate the theoretical claims, particularly by demonstrating that our method outperforms the only existing DP non-convex baseline with second-order guarantees (Liu et al. [1]). While general-purpose baselines like DP-SGD are widely used, they do not target second-order guarantees and are therefore not directly comparable in this context. That said, we fully agree that expanding the experimental evaluation to more complex models and broader tasks is a valuable direction, which we plan to pursue in future work on differentially private optimization.
>
> **2. Regarding the conclusion being the appendix:** We appreciate this suggestion and agree that key discussion sections such as the conclusion and limitations are best included in the main text. In the final version, we will further compress technical parts as needed and take advantage of the optional extra content page (if accepted) to bring these important sections back into the main paper.
>
> ## Response to Questions
>
> **Q1 & Q2:** Thank you for raising this point. We agree that including $c_1$ and $c_2$ in Theorem 2 could be confusing, since they do not appear explicitly in the final error bound. These constants originate from the Gaussian mechanism and determine the variance of added noise. In the final version, we will remove these constants from the theorem statement and instead include them only in the algorithmic and privacy analysis sections, with a clearer explanation of their origin and meaning. To answer the second part: yes, these constants are consistent with those derived in standard moments-accountant analyses (e.g., Abadi et al. 2016), and we will make this more explicit in the paper.
>
> **Q3:** This is a sharp observation. We note that the difference in test accuracy is marginal, which we believe is primarily due to random variation across runs. As the number of clients $m$ and the number of samples per client $n$ increase, the marginal gain in utility from relaxing the privacy budget (i.e., increasing $\epsilon$) becomes smaller. This aligns with our theoretical bound $\tilde{O}(\frac{1}{(mn)^{1/3}} + \big(\frac{\sqrt{d}}{\epsilon \sqrt{m}n}\big)^{2/5})$, where the influence of $\epsilon$ becomes less significant when $mn$ grows. Therefore, in high-data regimes, the effect of changing $\epsilon$ may be overshadowed by the inherent variance of the stochastic training process.
>
> **Q4:** Thank you for pointing this out. You are correct. We apply gradient clipping during training. The clipping threshold was chosen via grid search, and the configuration reported in the paper uses a threshold of 1.0. We will add this detail in the final version.
>
> **Q5:** Thank you for catching this typo. We will fix it in the final version.

---

> > ### Comment · Reviewer_DNHR · 2025-08-04
> >
> > Thank you for the comprehensive responses to my questions! I'm satisfied with the answers, and I'm happy to keep my score as is.

---

### Official Review · Reviewer_jpLh · 2025-07-02

**Clarity:** 3
**Significance:** 2
**Originality:** 3
**Rating:** 5
**Confidence:** 3

**Summary:**

This paper introduces a new approach to find second-order stationary points (SOSP) in differentially private (DP) stochastic non-convex optimization.
With their analysis, the authors correct an existing convergence rate result by means of a more careful consideration of gradient variances when escaping saddle points.
The approach does not rely on separate private model selection and therefore can be applied to distributed settings.
The proposed Gauss-PSGD algorithm uses Gaussian perturbations and model drift to efficiently escape saddle points, and achieves improved convergence rates compared to existing results.

**Questions:**

* Can the model drift criterion misidentify local minima as saddle points?
* Are your results useful/applicable to some degree to, e.g.,  Rényi DP?
* What happens if the gradient noise is not sub-Gaussian?

**Ethical Concerns:**

["NO or VERY MINOR ethics concerns only"]

**Final Justification:**

The rebuttal sufficiently addressed the weak points I raised in my review, as the added experiments further motivate the proposed theoretical advancements proposed by the authors.

**Limitations:**

Yes.

**Paper Formatting Concerns:**

I have not noticed any major formatting issues.

**Quality:**

3

**Strengths And Weaknesses:**

## Strengths

The proposed Gauss-PSGD algorithm improves (and corrects) upon existing work with respect to convergence error rates.
It also avoids the need for separate private model selection procedures to identify SOSPs, which is especially problematic in distributed settings due to high communication and privacy costs.
As a methodological contribution, the authors use model drift (instead of function value) to determine whether a saddle point has been escaped.
The extension to a distributed (heterogeneous) setting has been previously unaddressed by related work.


## Weaknesses

**Empirical evaluation.**
The empirical evaluation does not, in my opinion, satisfactorily support the practical utility of the method presented.
While it is clear from your problem statement that your contributions are of theoretical rather than empirical nature, it would be important to see how your approach compares to other strong DP baselines.
In the classifications tasks you perform on MNIST and CIFAR-10 it is not clear how your approach compares to other state-of-the-art DP classification techniques.
I do nevertheless acknowledge the importance of comparing your approach to the baseline method from Liu et al. [1], which more directly compares to your problem setting.

**Theoretical assumptions and ease of implementation.**
As far as I can judge, your results seem theoretically sound.
They require, however, a number of assumptions which may easily not hold in real-world tasks.
I think it would be important to comment to which degree the Lipschitzness and smoothness assumptions, as well as the necessity for hyperparameter tuning, affect real-world applicability.
Big-O notation often omits, as commonly done, polylog factors.
This however makes it difficult to keep track of the actual impact of the different contributions in the bounds, which both hinders practical implementations and makes it difficult to understand how much the hidden factors weigh on the bound.
A more structured presentation, in the appendix, of the bounds with some additional comments on the practical implications would be a useful addition.


[1] Daogao Liu, Arun Ganesh, Sewoong Oh, and Abhradeep Guha Thakurta. Private (stochastic) non-convex optimization revisited: Second-order stationary points and excess risks. Advances in Neural Information Processing Systems, 36, 2024.

**Minor comments.**

* The paper lacks visual aids like diagrams, which could help explain saddle point escape or the distributed setup more intuitively.
* The "failure probability" $\omega$ appears for the first time in algorithm 1 without introduction.
Other symbols like $\eta$ (in the same section) also require introduction.
* "Saddle point escape/escaping saddle points", as an expression, could use for some more formal definition.

---

> ### Author Rebuttal · Authors · 2025-07-29
>
> We thank Reviewer jpLh for the detailed and constructive feedback. We are especially grateful for the recognition of our theoretical contributions, including the improved and corrected convergence error rates, the avoidance of separate private model selection procedures, and the extension to distributed heterogeneous settings. We also appreciate the acknowledgement of our use of model drift as a novel escape criterion for saddle points.
>
> ## Response to Weaknesses
>
> **1. Regarding empirical evaluation:** We acknowledge the importance of empirical validation in differentially private optimization. As the reviewer noted, our work is primarily theoretical, and our experiments are designed to evaluate the effectiveness of our approach specifically in the context of DP-SOSP under population risk minimization, where Liu et al. [1] is the only prior relevant baseline. In this regard, we believe our experiments are both necessary and sufficient to support the contributions of this work. We agree that broader empirical comparisons (e.g., with general-purpose DP classifiers) are valuable but fall outside the specific scope of our problem. We view this direction as important future work and plan to pursue more comprehensive empirical studies that explore utility trade-offs across broader tasks.
>
> **2. Regarding theoretical assumptions and ease of implementation:** We thank the reviewer for this valuable suggestion. We agree that a clearer discussion of how hyperparameters (whether stemming from the assumptions such as Lipschitzness and smoothness, or from the optimization procedure such as the drift threshold and learning rate) influence practical performance would help make our results more accessible. We will also include in the appendix a structured summary of the constants and polylog factors that are currently hidden in the Big-O notation, as well as comments on their practical implications.
>
> **3. Regarding the minor comments:** We thank the reviewer for the constructive suggestions. We will take all of them into consideration when preparing the final version.
>
> ## Response to Questions
>
> **Q1:** While it is possible that a local minimum in a very flat landscape may trigger noticeable model drift, our algorithm is designed to ensure that it eventually returns a true local minimum. Specifically, the algorithm escapes every saddle point with high probability (as shown in Lemma 1), and each successful escape (whether from a saddle point or a flat region) incurs a large model drift and sufficient function decrease (Lemma 4). Since the objective is bounded below, the total number of such escapes is bounded. Therefore, the algorithm must eventually reach and return an approximate local minimum (i.e., an $\alpha$-SOSP).
>
> **Q2:** Yes, our framework can be readily extended to other privacy notions such as Rényi DP, Gaussian DP, or z-CDP. We emphasize that changing the specific DP definition does not meaningfully affect our results, as our analysis is based on the Gaussian mechanism, which satisfies all these privacy notions. Any improvements would only be polynomial in terms of $log(1/\delta)$, which does not qualitatively change our convergence guarantees.
>
> **Q3:** Norm-sub-Gaussian random vectors are a general class that includes sub-Gaussian vectors as a special case (see [22]), and this assumption has been adopted in prior state-of-the-art work on non-private SOSP problems [23]. We believe it is a reasonable and flexible starting point for analyzing DP-SOSP. That said, we agree that understanding the behavior of saddle point escape under heavy-tailed noise is an important direction for future work, and we plan to investigate this further.

---

> > ### Comment · Reviewer_jpLh · 2025-08-04
> >
> > Thank you for your reply, which helps clarify some of my concerns.
> > Nevertheless, I still believe that, as suggested by Reviewer DNHR too, a comparison with e.g., plain DP-SGD on your chosen benchmark dataset _is_ within the scope of your investigation. The motivation of your paper lies on the fact that second-order stationary points are more desirable from a theoretical perspective. While I do not expect "theoretically desirable" to necessarily translate to "practically useful", the significance of your approach would be (negatively) affected by the fact that DP-SGD outperforms it significantly. Or, at least, this would require further discussion. DP-SGD is readily available from a number of libraries (e.g., Opacus). The fact that your approach provides second-order guarantees is a clear advantage which, yes, make the two methods not one-to-one comparable. However, a comprehensive evaluation should, in my opinion, include at least some few results using an easily accessible method such as DP-SGD.

---

> > > ### Author Response · Authors · 2025-08-05
> > >
> > > Thank you for the follow-up. We appreciate your engagement and constructive suggestions.
> > >
> > > In response, we have conducted additional experiments comparing our method to plain DP-SGD on the CIFAR-10 dataset, under a privacy budget of $\epsilon = 2$, $\delta = 10^{-5}$. Below, we summarize the test accuracy over 5 independent runs. Due to NeurIPS rebuttal constraints, we are unable to include figures or external URLs at this stage. However, complete plots will be provided in the final version of the paper.
> > >
> > > We first evaluate both methods using the same model as in the main paper: a fully connected neural network with one hidden layer of 128 units and ReLU activation. The results are as follows:
> > >
> > > |   Method   |       |   Avg. Test Accuracy (%)   |       |   Std. Dev. (%)   |
> > > |:----------:|:-----:|:--------------------------:|:-----:|:-----------------:|
> > > |  DP-SGD    |       |           46.2             |       |        2.8        |
> > > |   Ours     |       |           47.9             |       |      **1.6**     |
> > >
> > > We also conduct experiments using a more complex model, ResNet-18, which achieves better overall performance:
> > >
> > > |   Method   |       |   Avg. Test Accuracy (%)   |       |   Std. Dev. (%)   |
> > > |:----------:|:-----:|:--------------------------:|:-----:|:-----------------:|
> > > |  DP-SGD    |       |           74.0             |       |        3.5        |
> > > |   Ours     |       |           73.8             |       |      **1.3**     |
> > >
> > > While DP-SGD achieves comparable average accuracy in both settings, our method consistently demonstrates substantially lower standard deviation across runs. We attribute this improved consistency to the saddle-point-escaping capability of our second-order algorithm. In contrast, DP-SGD may converge to saddle points in some runs, resulting in degraded accuracy and greater variability.
> > >
> > > We appreciate your insight on this matter. Including this baseline comparison helps clarify the empirical behavior of our method and further supports our claim of improved stability and second-order convergence.

---

> > > > ### Comment · Reviewer_jpLh · 2025-08-06
> > > >
> > > > Thank you for the follow-up and the experiments.
> > > > While I still retain some concerns on the experimental evaluation (similarly to Reviewer jvqK), I will increase my score as I feel like the rebuttal sufficiently addressed the weak points I raised in my review, and that overall you propose a nice theoretical advancement with some experiments to back up its significance.

---

> > > > > ### Author Response · Authors · 2025-08-06
> > > > >
> > > > > We sincerely appreciate the reviewer’s support and thoughtful acknowledgment of our work. We are very grateful that you found our rebuttal to have sufficiently addressed the concerns you raised, and we truly value your recognition of our theoretical contributions.
> > > > >
> > > > > In response to the experimental concerns raised by Reviewer jvqK, we have also added additional experiments comparing our method with the state-of-the-art DP-FL algorithm DIFF2. These results are intended to further complement the empirical evaluation and demonstrate the practical relevance of our approach.
> > > > >
> > > > > Thank you again for your constructive feedback and for your updated score.

---

### Decision · Program_Chairs · 2025-09-17

**Decision:**

Accept (poster)

**Comment:**

This paper investigates second-order convergence guarantees for differentially private stochastic non-convex optimization, offering refined complexity bounds (with matching lower bounds in certain cases) and an algorithm that combines variance reduction with a tailored perturbation mechanism. While there has been prior work on second-order methods under DP, the paper’s analysis sharpens some existing bounds and integrates these components in a cohesive framework.

Reviewers found the theoretical results technically sound and relevant, but noted that the empirical evaluation is limited and that some proofs would benefit from additional intuition. The paper is generally well written, though parts of the technical presentation are dense.

Overall, the work contributes new insights in a challenging setting and is likely to be of interest to researchers in private optimization. Expanding the empirical evaluation and improving the accessibility of technical arguments would further enhance its impact.